# Cell-Free Fetal DNA Screening Analysis in Korean Pregnant Women: Six Years of Experience and a Retrospective Study of 9327 Patients Analyzed from 2017 to 2022

**DOI:** 10.3390/jpm13101468

**Published:** 2023-10-06

**Authors:** Ji Eun Park, Kyung Min Kang, Hyunjin Kim, Hee Yeon Jang, Minyeon Go, So Hyun Yang, Daeun Jeong, Hyeonmin Jeong, Jong Chul Kim, Seo Young Lim, Dong Hyun Cha, Sung Han Shim

**Affiliations:** 1Center for Genome Diagnostics, CHA Biotech Inc., Seoul 06125, Republic of Korea; ditto6626@chamc.co.kr (J.E.P.); rkdrudals@chamc.co.kr (K.M.K.); hjkim0530@chamc.co.kr (H.K.); jhyeon@chamc.co.kr (H.Y.J.); raculase@chamc.co.kr (M.G.); xceanftb@chamc.co.kr (S.H.Y.); jckim43@chamc.co.kr (J.C.K.); 2Department of Biomedical Science, College of Life Science, CHA University, Seongnam 13488, Republic of Korea; grace.eunj@gmail.com (D.J.); rice1811m@gmail.com (H.J.); chagene@chamc.co.kr (S.Y.L.); 3Department of Obstetrics and Gynecology, CHA Gangnam Medical Center, CHA University, Seoul 06125, Republic of Korea

**Keywords:** cell-free DNA (cfDNA), non-invasive prenatal test (NIPT), body mass index (BMI), fetal fraction screening test

## Abstract

Cell-free DNA (cfDNA) screening for normal fetal aneuploidy has been widely adopted worldwide due to its convenience, non-invasiveness, and high positive predictive rate. We retrospectively evaluated 9327 Korean women with single pregnancies who underwent a non-invasive prenatal test (NIPT) to investigate how various factors such as maternal weight, age, and the method of conception affect the fetal fraction (FF). The average FF was 9.15 ± 3.31%, which decreased significantly as the maternal body mass index (BMI) increased (*p* < 0.001). The highly obese group showed a ‘no-call’ rate of 8.01%, which is higher than that of the normal weight group (0.33%). The FF was 8.74 ± 3.20% when mothers were in their 40s, and lower than that when in their 30s (9.23 ± 3.34, *p* < 0.001) and in the natural pregnancy group (9.31% ± 3.33). The FF of male fetuses was observed to be approximately 2.76% higher on average than that of female fetuses. As the gestational age increased, there was no significant increase in the fraction of fetuses up to 21 weeks compared to that at 10–12 weeks, and a significant increase was observed in the case of 21 weeks or more. The FFs in the NIPT high-risk result group compared to that in the low-risk group were not significantly different (*p* = 0.62). In conclusion, BMI was the factor most associated with the fetal fraction. Although the NIPT is a highly prevalent method in prenatal analysis, factors affecting the fetal fraction should be thoroughly analyzed to obtain more accurate results.

## 1. Introduction

Numerical and structural chromosomal abnormalities are the most common causes of prenatal and birth-related defects. The American College of Obstetricians and Gynecologists (ACOG) guidelines recommend that all pregnant women be offered options for prenatal screening and diagnostic testing [1]. To this date, a commonly used method to assess fetal development during pregnancy is non-invasive screening tests conducted in the first and second trimesters. These tests include measuring nuchal translucency (NT) thickness, pregnancy-associated plasma protein-A (PAPP-a), serum α-fetoprotein (AFP), unconjugated estriol (uE3), performed in the first trimester, and measuring beta-human chorionic gonadotropin (β-hCG) levels, performed in the second trimester [2]. Although this combined screening test has a positive predictive value (PPV) of approximately 95–97%, 3–5% is still confirmed as a false positive [3]. Another approach to test for chromosomal abnormalities in the fetus is invasive tests such as amniocentesis and chorionic villus sampling (CVS). Although these tests have been proven safe when performed by a trained operator, they are not without risks, including bleeding, infection, premature birth, miscarriage, stillbirth, fetal damage, and an excess abortion rate of 0.5% to 1% [4,5].

Fan et al. described a groundbreaking achievement in DNA sequencing. They successfully applied a technique called massive parallel sequencing (MPS) to sequence cell-free DNA extracted from maternal plasma [6]. Subsequent studies further validated the non-invasive prenatal test (NIPT) as a reliable tool, confirming its effectiveness and practicality. Since its initial success, the NIPT has been integrated into routine clinical practice since 2011, primarily as a prenatal screening tool for the detection of trisomy 21 (Down syndrome). Over time, the scope of the NIPT has expanded to include the detection of other aneuploidies, such as trisomy 18, trisomy 13, and abnormalities in sex chromosomes. This broader application of the NIPT has allowed for a more comprehensive screening approach for prenatal care [7].

Currently, the NIPT using cell-free fetal DNA (cfDNA) in circulating maternal blood is the most widely used screening method for chromosomal anomalies [8]. It is estimated that between 4 and 6 million pregnant women undergo a NIPT annually worldwide, and this number is expected to surpass 15 million within a decade [9,10]. Compared with other screening tests, the NIPT has shown a lower false positive rate (FRP) and higher positive predicts value (PPV) [11].

The median fetal DNA fraction in DNA obtained from maternal blood at 11–13 weeks of gestation is around 10%, which is sufficient for the NIPT; however, approximately 3–5% of the total cases result in a “no-call”, which means that analysis is not possible because the fetal fraction (FF) is below 4% [12,13,14]. The fetal DNA fraction is associated with numerous factors related to the characteristics of the mother and fetus, such as IVF pregnancy, maternal body mass index (BMI), maternal drug exposure, early gestational age (GA), and certain racial backgrounds [13,15,16,17].

Therefore, in this retrospective study, we aimed to investigate which factors affected the NIPT analysis of approximately 10,000 Korean mothers over 6 years in an attempt to reduce the ‘no-call’ result.

## 2. Materials and Methods

### 2.1. Subject

We conducted a retrospective cohort study of 9327 women who underwent aneuploidy screening at the CHA Medical Center between July 2017 and December 2022. To ensure a homogeneous study population, we included only women with single pregnancies and excluded those with twin or triple pregnancies. In addition, when requesting a test, we were notified of vanishing twins, or samples with Y chromosome results of less than 1.5%, which were excluded from the analysis because vanishing could not be ruled out. The study population had a mean maternal age of 36.3 years (interquartile range, IQR: 34–39 years) and a mean gestational age of 12.37 weeks (IQR: 11.6–12.4 weeks). Table 1 summarizes the characteristics of the study population.

### 2.2. NIPT Analysis

Approximately 10 mL of peripheral blood was collected from each participant into a cell-free DNA BCT tube (Streck, Omaha, NE, USA). Blood samples were centrifuged at 1200× *g* for 10 min at 4 °C. The plasma portion of the blood was centrifuged for 10 min. at 4 °C, and the cfDNA was extracted from 1 mL of plasma using QIAamp Circulating Nucleic Acid Kit (Qiagen, Hilden, Germany). All NIPTs were performed at the Center for Genome Diagnostics CHA Biotech Inc. using Ion S5XL sequencing systems in accordance with the manufacturer’s protocol (Thermo Fisher, Waltham, CA, USA). Fourteen cfDNA samples were loaded onto Ion 540 Chip Kit (version 2.0; Life Technologies, Carlsbad, CA, USA). The raw reads of each sample were more than 5 million, and the rate of uniquely mapped reads was more than 65.0%. Ion 540^TM^ Chip Kit was used to yield an average 0.3× sequencing coverage depth per nucleotide. We used massive parallel sequencing to analyze cell-free fetal DNA in the maternal plasma and measured the relative gains or losses in chromosomal representations to detect chromosomal abnormalities.

Body mass index (BMI) was divided into five categories in accordance with the Asia–Pacific cutoff points recommended by the World Health Organization (WHO) in 2017; <18.5 kg/m^2^ was considered underweight, 18.5 to 22.9 kg/m^2^ was normal, 23.0 to 24.9 kg/m^2^ was overweight, 25.0 to 30.0 kg/m^2^ was obese, and >30 kg/m^2^ was considered highly obese. The presence or absence of the Y chromosome determines fetal sex. If a Y chromosome-specific gene was present, the individual was judged to be male; otherwise, they were considered to be female.

### 2.3. Additional Diagnostic Tests

In the NIPT, women who were revealed to be at high or critical risk for fetal chromosomal abnormalities were subjected to amniocentesis or chorionic villus sampling (CVS), and additional tests such as karyotype analysis, multiplex ligation-dependent probe amplification (MLPA), and array comparative genomic hybridization (aCGH) were performed. For karyotype analysis, the amniotic fluid was obtained via amniocentesis or CVS. Amniotic fluid cells in 4.5 mL of BIO-AMF^TM^ medium (Biological Industries, Cromwell, CT, USA) were cultured in a 37 °C incubator with 5% carbon dioxide. Cells were harvested after 10–12 d. An all-chromosome image analysis system based on “An International System for Human Cytogenetic Nomenclature, ISCN2020” was used. The Affymetrix CytoScan Optima array platform was used for the aCGH. DNA was extracted from amniotic fluid or chorionic villi after invasive prenatal diagnostics (Kurabo Industries Ltd., Osaka, Japan). The G-CMA Database (GC Genome, Yong-in, Republic of Korea) was used for the data analysis. Copy number variations (CNVs) were classified using Online Mendelian Inheritance in Man (OMIM), Database of Genome Variants (DGV) and Decipher. Pathogenic, likely pathogenic, variant of uncertain significance (VUS), likely benign, and benign categories were used for CNV allocation. For VUS, aCGH was performed on parents to verify whether or not the CNVs were inherited from parents with a normal phenotype. Inherited CNVs from parents with normal phenotypes were considered benign. Multiplex ligation-dependent probe amplification (MLPA) was performed in accordance with the manufacturer’s instructions. The genomic DNA was denatured and hybridized with P070 probes at 60 °C for approximately 16 h. PCR amplification was performed after 15 min of ligation at 54 °C, using Cy5 labeled primers. Fluorescent amplification products were separated based on their lengths via capillary electrophoresis in ABI3500DX Genetic Analysis System (Thermo Fisher Co., MA, USA), and the results were analyzed using GeneMarker version 3.1 software. (Softgenetics, State College, PA, USA). The probe ratios of deletion and duplication were fixed at 0.75 and 1.25, respectively.

### 2.4. Statistical Analysis

Normally distributed data are expressed as mean and standard deviation (SD), with median and interquartile range (IQR) for non-normally distributed data for continuous variables. Categorical variables are presented as absolute values and percentages. A *t*-test was conducted using the Lawstat R package (v4.3.1) to calculate statistical significance. Statistical significance was set at *p* ≤ 0.05.

## 3. Results

In total, 9327 pregnant women participated in this study, of which 3655 conceived through assisted reproductive technology (ART) and the remaining 5672 conceived naturally. Mean maternal age, weight, height, and BMI were 36.3 (range 23 to 47 years), 58.8 (interquartile range, IQR, 52.4–63.4 kg), 162.47 (interquartile range, IQR, 159.0–166.0 cm), and 22.3 (interquartile range, IQR, 19.9–23.8 kg/m^2^), respectively. NIPTs were performed for a mean GA of 12 weeks and 4 days, ranging from 9 weeks to 31 weeks. Most tests were performed at 11–13 weeks (approximately 12 weeks) (Figure 1A), and most subjects had a BMI of 18.0–23.9 (Figure 1B). Of these women, 4704 were predicted to conceive males and 4582 were predicted to conceive females. The mean FF of total group was 9.15% (Figure 2A). The highest FF was 28.2% in 4704 male fetuses, while the highest FF among 4582 female fetuses was 15.2% (Figure 2B ).

We determined whether or not 4.5% of the FF were at borderline low risk. Even if the FF was higher than the borderline, raw data (total bases, key signal, total reads, read length, etc.), unique reads (more than two million), z-scores (−2.5 < z-score < +2.5), and GC contents (38.5 to 42.5) were considered to decide whether to resample or not. Due to variations in the FF, it was difficult to accurately estimate the precise amount of increase. Therefore, we conducted resampling based on specific criteria, considering the initial FF and pregnancy method. If the FF was between 3.5–4.0%, resampling was scheduled after 3 weeks. If the FF exceeded 4.0%, it was scheduled after 2 weeks. For pregnancies assisted by ART, an additional week was included.

Results were obtained from 9013 of the first samples out of a total of 9327 samples. Of the 314 samples that failed in the first attempt, 16 were immediately recollected via hemolysis, 42 were subjected to alternative testing without performing the NIPT again, and three were intrauterine fetal deaths (IUFD) before the test results were available. Subsequently, 253 individuals underwent a second round of testing. Among them, 35 samples received a conclusive result, indicating ‘no-call’. In the alternative test group, resampling group, and final no-call group, the FF was significantly decreased, and BMI was significantly increased (*p* < 0.001) compared to those in the first test passed group. There was no significant relationship between the IUFD and hemolysis groups. For 253 specimens, except for the group resampled due to hemolysis, re-collection was completed in approximately 19.4-day intervals. After retesting, the FF increased by approximately 1.55%, reaching 5.95%. The mean BMI of the 80 subjects with no-call results was 27.51 ± 5.09, which was higher than the 22.15 ± 3.24 BMI in the group with only one trial. Although it had a cut-off value, the mean BMI of the retested group was 24.71 ± 3.07, which was significantly higher than that of the first test passed group (Table 2).

For each BMI group, the mean values were 10.45%, 9.69%, 8.55%, 7.65%, and 6.26% in the underweight, normal weight, overweight, obese, and highly obese groups, respectively. BMI was negatively correlated with FF. Compared to that in the normal group, the mean value of the FF was significantly higher in the underweight group and significantly lower in the overweight, obese, and highly obese groups. In the cases of male and female fetuses, the mean FFs were 10.50 ± 3.77% and 7.74 ± 1.94%, respectively. In all BMI groups, the fetal fraction was significantly higher in male fetuses than in female fetuses (*p* < 0.001; highly obese group, *p* < 0.05) (Table 3).

Overall, 314 of the 9327 cases in the first sampling had no-call results. This accounted for 3.37% of the total sample. There were 243 cases of low fetal cfDNA, 55 cases of a fluctuated z-score (If the Z score is 2.5 or higher or −2.5 or lower), and 16 cases of hemolysis, with proportions of 2.61%, 0.59%, and 0.17%, respectively. Due to insufficient fetal cfDNA and because the samples did not reach the thresholds for quality control, the mean FF of the resampling group was 3.68%, and the BMI was 25.91. In the normal BMI group, the reasons for resampling were evenly distributed and included fragmented DNA, hemolysis, and a low FF. However, in the group with a higher maternal BMI, the proportion of low FFs among the total reasons for resampling increased gradually. Of the 314 patients with a low FF and fluctuated z-score, 45 immediately underwent invasive or serological testing, with 234 resamplings. However, even in the retest group, 35 patients did not exceed 4.5% of the FF; therefore, an alternative examination method (integrated/quadruple test or CVS/amniocentesis) was performed. Among the patients, 61 were overweight and 25 had a BMI of 30 or higher (Table 4).

In addition, 9.31% ± 3.33 of natural pregnancies and 8.73% ± 3.23 of ART pregnancies were significantly higher than the percentage of artificial insemination in terms of mean FF (*p* < 0.001). The difference in the FF based on the number of weeks of gestation had no significant effect until 21 weeks of gestation. However, in samples collected at a gestational age of 21 weeks or more, the means were 3.08% for female and 7.73% for male fetuses, respectively. This number was significantly higher than that in the samples collected between 11 and 12 weeks of gestation. (Table 5).

When comparing the FF according to maternal age with that of pregnant women in their 30s (9.23 ± 3.34), the FF of women in their 20s was 9.50 ± 3.31, which was not significant (*p* = 0.13), and was 8.74 ± 3.20 for women in their 40s, which showed significantly lower results than those in their 30s (*p* < 0.001) (Table 6).

Briefly, 37 of the 80 patients for whom the NIPT results failed were confirmed using invasive methods such as CVS or amniocentesis. Among the 37 patients, 1 who received a no-call result due to a fluctuating z-score obtained an abnormal result of 46,00,der?(14). The rest were confirmed to have normal karyotypes. Another subject in the no-call group was 12 weeks and 5 days pregnant at the time of the NIPT, had a normal BMI, and no specific problems other than an IVF pregnancy. During the two NIPTs, 3.23% and 3.32% of the FFs were present, but <4.5%, resulted in no-calls. Subsequently, amniocentesis confirmed that the fetal chromosomes were normal. However, the patient was transferred to another hospital after being diagnosed with acute myeloid leukemia (AML) around the 30th week of pregnancy. As the patient was transferred, the condition could not be followed-up. Seven mothers underwent a serum screening test together, and two of them showed values of 2.725 and 2.90 for neural tube defects, but a low-risk result was obtained in the AFP test. However, this was not confirmed until birth. Moreover, three subjects were stillborn during the test, all of which were males. One of the patients was confirmed to have no chromosomal abnormality through the abortus MLPA test, and trisomy 18 (FF4.28%, Z-score 5.13) was suspected in one patient based on the NIPT results; however, the pregnancy was terminated, and therefore, further tests were not possible. Other factors were not tested in this study. Thirty-three patients who did not want to be transferred to another institution or undergo any further examination were lost to follow-up (Table 7).

High-risk results were obtained for 165 of the 9327 samples. The mean FF of the high-risk group was 9.34% + 3.27, and there was no statistical significance when compared to that of 9.19% + 3.27 of the low-risk group (*p* = 0.56). Among all high-risk groups, trisomies 21, 18, and 13 accounted for the largest proportion (51.5%), and 83 patients were followed up. In the case of trisomy 21, additional tests were performed on 46 of 56 high-risk patients, and concordant results were obtained in 97.8% of cases. In a patient with a false positive result, mosaicism was observed in the CVS test (mos 47,XX,+21[11]/46,XX[39]); however, placenta-restricted mosaicism was suspected because the amniocentesis result was normal. Therefore, placenta-restricted mosaicism was confirmed via securing the placenta during childbirth [18]. Invasive methods for chromosomes 18 and 13 showed PPVs of 47.4% and 14.3%, respectively. For autosomal abnormalities, the same result (PPV: 6.3%) was observed in only one case. In 41 cases wherein sex chromosome abnormalities, such as Turner syndrome, Klinefelter syndrome, Triple X, and XXY, were suspected, karyotyping or CMA (chromosomal microarray analysis) was performed in 34 cases, and the same results as those from the NIPT were confirmed in 15 cases (PPV, 44.1%). Chromosomal aberrations thought to originate from the mother were confirmed via the mother’s own mutation or chromosomal abnormalities in 13 of the 19 cases. Among the 13 cases, the mother’s sex chromosome was 47,XXX in four cases, and Turner mosaicism was confirmed in one case (mos 45, X[34]/46, XX[66]). The remaining eight cases were identified through CMA or fragment analysis as partial deletions or duplications in the autosomes. Irrespective of whether the mother herself was confirmed or not, fetal examinations were performed in 9 out of 19 cases, 5 fetuses showed normal results, and 4 cases confirmed results that were consistent with those of the mother (Table 8).

## 4. Discussion

The NIPT uses fetal cfDNA in maternal blood to detect fetal aneuploidies before birth and thus, is safer relative to other invasive prenatal tests [19]. The fetal fraction, defined as the ratio of fetal cfDNA to total cfDNA in the maternal plasma, is an important parameter for the reliability of NIPT results [20]. Various factors such as maternal weight, method of conception, gestational age, drug exposure, genetic condition, and errors in experimental procedures can affect the FF [21].

In our six years of experience, we have encountered several factors that affect the FF. As maternal BMI and age increase, the FF is known to decrease during IVF pregnancy. Moreover, a higher FF has been reported in male fetuses than in female fetuses. Our results are consistent with this finding.

Maternal obesity and IVF pregnancies are inversely proportional to the FF [13]. Previous studies conducted a multivariable regression analysis to confirm the relationship between fetal fraction and maternal weight. According to their calculation, when the average maternal weight is 60 kg, the FF usually appears to be 11.7%; however, when the average maternal weight is 160 kg, the FF decreases to 3.9%. Furthermore, the proportion of ‘no-call’ cases in the same population increased from 0.7% to 51.1% for the same compared population [22]. One of the causes of NIPT failure is in vitro fertilization. From this type of fertilization, the number of cases with a FF of less than 4% was shown to have a failure rate that was 3.8 times higher than that of natural pregnancies. With each additional year of maternal age and kilogram of maternal weight, the failure rate increases by 2% and 5%, respectively [23]. In our study, the differences in the FF according to the pregnancy method were consistent with the findings of other studies demonstrating a reduced FF for single pregnancies after ART compared to that for natural conception [24]. Elena et al. reported that in a cohort of 12,000 mothers, the FF in male fetuses was 9.9% and the FF in female fetuses was 6.8%, indicating that the FF in male fetuses was higher [25]. In our study, the male FF was 2.76% higher than was the female FF.

In a study by Kinnings et al., FFs increased by 0.44%, 0.083%, and 0.821% per week at 10–12.5 weeks, 12–20 weeks, and 20 weeks of gestation, respectively [17]. The rate of increase in the FF did not constantly increase as the GA did. However, it has also been reported that FF increases significantly with gestational age increases [25]. In our results, there was no significant increase in the FF until 21 weeks of gestation compared with the FF at 11–12 weeks of gestation.

A major advantage of the NIPT is its high sensitivity and specificity for common aneuploidies (T21, T18, and T13). Several validation studies have reported high sensitivity (98.6–100%) and specificity (99.7–100%) for T21 in various populations [26,27]. Our results also showed high sensitivity and specificity for T21, T18, T13, and SCA, and an excellent PPV for T21. Thus, it is a superior test for detecting T21, which is not easily confirmed via ultrasound, compared with T18 and T13, which are relatively easy to detect using soft markers on ultrasound. Several factors contribute to the low false positive and false negative results of NIPT compared to those of conventional prenatal screening tests. Low FFs, maternal CNVs, and fetal/placental mosaicism are some of these factors [28]. False negative results are rare in NIPT, with a frequency of 0.08% [29]. Likewise, in our study, there were no false negatives for the common trisomy.

Confined placental mosaicism (CPM) occurs in 1–2% of pregnancies and is known to be more likely in trisomy 13 and monosomy X than in trisomy 21 or 18 [30]. In our study, the false positive rate was high for T13 and monosomy X. Most cfDNA test methods assume that the mother has a normal karyotype. However, there are cases wherein maternal abnormalities are not recognized (for example, when the mother’s chromosome is 47,XXX); as maternal age increases, cells with some abnormal chromosomes exist in a small proportion in the form of mosaicism, which can lead to false positive results [31].

The main reason for no-calls in the NIPT group was maternal BMI. Therefore, in patients with a BMI of 30 or higher, other tests should be performed together in preparation for a high no-call ratio. In addition, conducting repeated blood draws while waiting for a later gestational age is an unreliable approach for overcoming the low FF in subjects with a higher BMI and at earlier gestational weeks. As shown in Table 4, even if the normal BMI group had a no-call result after resampling, the no-call ratio was 0.33%, whereas that of the high-obesity group remained at 8.01%. In a study by Rolnik et al. in which NIPT was performed using two methods, platform A analysis (n = 8583) showed a no-call (FF below 2%) rate of 0.7% in the normal group, 5.5% in the BMI 30.0–34.9 group, and 14.7% in the BMI 35.0–39.9 group. In platform B (n = 5640), the normal group had a no-call (FF below 2%) rate of 0.4%, but the severely obese group with a BMI of 30.0 to 34.9 and 35.0 to 39.9 showed a no-call rate of 1.7% and 7.1%, respectively. They stated that for women with a BMI of 35 or higher, the increase in the fetal fraction is minimal with an increasing gestational age, and that delaying testing is unlikely to reduce the test failure rate in this population [32]. In such cases, patients should undergo invasive tests such as amniocentesis or CVS. For obese women, if the test findings are confirmed to be normal after the first NT measurement, performing first-trimester screening together with NIPT will reduce the need for invasive procedures.

Resampling is not only a burden to the experimenter, but also increases the anxiety of pregnant women about the stability of pregnancy. Therefore, rather than sampling blood in the same week of gestation for all patients, performing the test considering the patient’s current condition and the week of resampling may reduce the costs and efforts required for testing and reassure the pregnant woman. The American College of Medical Genetics and Genomics showed that when a pregnant woman underwent blood sampling in the appropriate week of gestation, it resulted in a low FF, and repeated sampling was not recommended [1]. However, the patient could be offered an alternative screening test or repeated sampling for NIPT if the ultrasound is normal and the individual wishes to avoid an invasive diagnosis. Caldwell et al. (2021) reported success in obtaining 593 (84.2%) of 704 pregnant women through resampling. Because no reportable results were obtained in the first trial, invasive tests had to be avoided [33].

Despite having a normal BMI, if there is no increase in the FF even after resampling, close monitoring is required due to maternal health concerns or the possibility of fetal abnormalities. In a large study of over 16,000 patients with a failure rate of 3%, the prevalence of aneuploidy in the overall cohort was 0.4%. In contrast, the no-call group showed a higher prevalence, 2.7% [11]. In our study, one patient in the normal BMI group was diagnosed with acute leukemia at approximately 30 weeks of gestation. Thus, maternal blood disorders presumably affect the FF.

## 5. Conclusions

Our study found that the FF was significantly influenced by the maternal BMI, age, fetal sex, and method of conception. If the initial NIPT does not produce a FF greater than 4.5%, we recommend resampling approximately 3–4 weeks before considering invasive testing. Patients with severe obesity are at a higher risk of obtaining a “no-call” result; therefore, it is recommended that patients fully discuss the potential need for additional testing prior to undergoing a NIPT. The NIPT is carried out on the premise that the mother has a normal karyotype, and it is possible to detect chromosomal abnormalities in the mother by chance; the contents, including the possibility of confirming the mother’s chromosomal abnormality, should be reviewed and understood before the test. It is crucial to observe the pregnancy even if there is no difference between the first and resampling NIPT results, even when the pregnancy is progressing without a special high risk. Overall, our findings highlight the importance of considering the various factors that may affect the FF in the clinical application of NIPT.

## Figures and Tables

**Figure 1 jpm-13-01468-f001:**
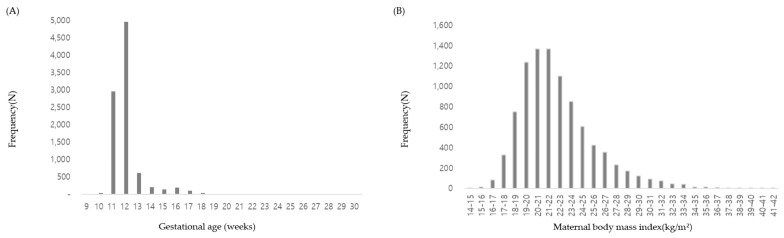
Distribution of gestational age and maternal body mass index. (**A**) Of the total 9327 pregnant women, 92.0% (n = 8581) underwent a NIPT in their first trimester of pregnancy. (**B**) The majority (71.5%, n = 6673) of women had a BMI ranging from 18.0 kg/m^2^ to 23.9 kg/m^2^.

**Figure 2 jpm-13-01468-f002:**
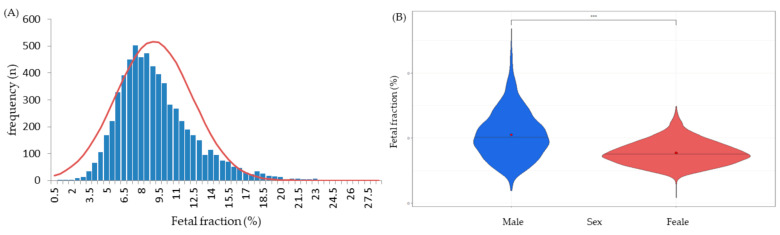
(**A**) The frequency distribution of FF of all subjects (n = 9327) showed a normal distribution (red line). (**B**) Frequency distribution of the FF in male (blue, n = 4704) and female (red, n = 4582) fetuses. *** *p* values are for the comparison between male and female group (*p* < 0.001).

**Table 1 jpm-13-01468-t001:** Maternal and pregnancy characteristics of a singleton population.

Characteristic	Results
Sample size	9327
Maternal age (y), mean ± SD	36.3 ± 3.63 (range 23 to 47 years)
Maternal weight (kg), mean ± SD (IQR) *	58.8 ± 9.27 (52.4, 63.4)
Maternal height (cm), mean ± SD (IQR)	162.5 ± 5.08 (159.0, 166.0)
Body mass index (kg/m^2^), mean ± SD (IQR)	22.3 ± 3.33 (19.9, 23.8)
Method of conception (%)	
Spontaneous	5672 (60.81)
In vitro fertilization	3655 (39.19)
Gestational age (weeks and days), mean (IQR)	12 weeks 3.7 day (11 w 3 d, 12 w 4 d)
Fetus gender (%) **	
Male	4704 (50.43)
Female	4582 (49.13)
Inability to conclude on sex chromosomes	41 (0.44)

* IQR, interquartile range, ** prediction based on Y-chromosome-specific genes.

**Table 2 jpm-13-01468-t002:** Overall efficiency of NIPT diagnosis in our laboratory based on solely FFs.

Group	Results	Subjects (n)	Portion (%)	Fetal Fraction (Mean % ± SD, *p* Value *)	BMI (Mean % ± SD, *p* Value *)
Passed test	1st test passed	9013	96.63	9.31 ± 3.23	22.15 ± 3.24
	2nd test passed (resample and retest)	218	2.34	4.48 ± 1.86, <0.001	24.71 ± 3.70, <0.001
	Hemolysis	16	0.17	9.50 ± 5.60, 0.89	22.16 ± 3.19, 0.98
No-call	Replace other diagnosis	42	0.45	3.26 ± 0.93, <0.001	28.03 ± 4.98, <0.001
	2nd test fail (no-call)	35	0.38	3.85 ± 1.02, <0.001	27.48 ± 4.99, <0.001
	IUFD ** before results	3	0.03	6.44 ± 3.96, 0.413	20.60 ± 1.56, 0.296

* *p* values are for the comparison between the fetal fraction of the 1st test passed group and that of others. ** IUFD, intrauterine fetal death.

**Table 3 jpm-13-01468-t003:** Fetal fraction according to the World Health Organization (WHO) body mass index criteria for the Asia–Pacific region.

BMI Group	Sample (n)	Fetal Fraction (Mean% ± SD)				*p* Value *
		Total	Male	Female	Not Determined for Sex **	
Underweight	714	10.45 ± 3.53	12.32 ±3.81	8.61 ± 3.81	8.95 ± 1.96	<0.001
Normal	5400	9.71 ± 3.28	11.24 ± 3.61	8.08 ± 1.90	11.84 ± 4.90	
Overweight	1571	8.60 ± 2.82	9.66 ± 3.35	7.42 ± 1.72	9.26 ± 2.37	<0.001
Obese	1330	7.76 ± 2.81	8.49 ± 3.44	6.68 ± 1.69	8.51 ± 2.56	<0.001
Highly obese	312	6.37 ± 2.05	6.58 ± 2.49	5.97 ± 1.58	-	<0.05

* *p* values for the comparison between the total fetal fraction of the normal group and the others. ** The high-risk group of sex chromosomal abnormalities in fetuses such as Klinefelter syndrome, Turner syndrome, and triple X.

**Table 4 jpm-13-01468-t004:** Reasons for resampling and results of retest according to BMI group.

Reason for No-Call	1st Diagnosis Failed (n, %)	2nd Diagnosis (n, % *)		
		Alternative Test	Resample	No-Call
Underweight, n = 714	10 (0.01) *		9 (1.26)	1 (0.14)
Low FF	4 (0.56) **		4	1
Fluctuated z-score	4 (0.56) **		3	
Hemolysis	2 (0.28) **		2	
Normal, n = 5400	95 (0.06) *	12 (0.22)	77 (1.43)	6 (0.11)
Low FF	63 (3.22) **	10	49	4
Fluctuated z-score	23 (8.82) **	2	19	2
Hemolysis	9 (1.26) **		9	
Overweight, n = 1571	52 (0.04) **	4 (0.25)	44 (2.80)	4 (0.25)
Low FF	42 (5.88) **	4	34	4
Fluctuated z-score	10 (1.40) **		10	
Obese, n =1330	111 (0.14) *	15 (1.13)	83 (6.24)	13 (0.98)
Low FF	92 (12.89) **	14	67	11
Fluctuated z-score	14 (1.96) **	1	11	2
Hemolysis	5 (0.70) **		5	
Highly obese, n =312	46 (6.44) *	14(4.49)	21(6.73)	11 (3.53)
Low FF	42 (5.88) **	14	18	10
Fluctuated z-score	4 (0.56) **		3	1
Sum	314 (3.37) *	45(0.48) *	234(2.51) *	35(0.38) *

* Percentage of total of 9327 samples; ** percentage of total for each group.

**Table 5 jpm-13-01468-t005:** Fetal fraction according to gestational age (GA).

GA Group	Sample (n)	Fetal Fraction (%) (Mean ± SD)	*p* Value *
<10	52	9.02 ± 3.33	0.77
11–12	7912	9.15 ± 3.27	
13–14	828	8.97 ± 3.38	0.14
15–16	343	9.19 ± 3.71	0.80
17–18	155	9.51 ± 3.28	0.18
19–20	12	8.47 ± 2.52	0.47
>21	25	17.16 ± 2.29	<0.05

* *p* values are for comparison between the 11 and 12 GA and other groups.

**Table 6 jpm-13-01468-t006:** Fetal fraction by maternal age group.

Maternal Age (Years)	Samples (n)	Portion (%)	Fetal Fraction (Mean %, SD)	*p* Value *
20–29	359	3.8	9.50 ± 3.31	0.13
30–39	7201	77.2	9.23 ± 3.34	
40–49	1767	18.9	8.74 ± 3.20	<0.001

* *p* values are for the comparison between the 30–39 years group and others.

**Table 7 jpm-13-01468-t007:** Additional examination and follow-up of 80 pregnant women with a final no-call decision at NIPT.

Alternative Test	Cases	Follow-Up
Invasive test (Amniocentesis or CVS)	37	36 of 37 were identified as normal diploid, but one fetus was identified as 46,00,der?(14) via amniocentesis. A patient with normal amniocentesis was diagnosed with acute myeloid leukemia (AML) in the third trimester of pregnancy.
Non-invasive test (Quadruple or integrated test)	7	5 out of 7 women received a low-risk result in the serum test, but 2 patients obtained scores of 2.725 and 2.90, respectively in the neural tube defect risk test. These two patients were judged as low-risk group via the AFP test.
Follow-up loss	33	Thirty-three pregnant women either transferred to another institution for examination or refused further examination.
IUFD	3	Three pregnant women underwent IUFD at 11 w + 5 d, 12 w, and 12 w + 6 d, during the NIPT or after determining the no-call result. Three cases had a fetal fraction of 3.04%, 4.28%, and 12.0%, respectively.

**Table 8 jpm-13-01468-t008:** Additional follow-up results for 165 abnormal results from the NIPT.

	High Risk	True Positive	False Positive	Follow-Up Loss	PPV (%)
Trisomy 21	56	45	1 *	10	97.8
Trisomy 18	21	9	10	2	47.4
Trisomy 13	8	1	6	1	14.3
Autosomal abnormality	20	1	15	4	6.3
Sex chromosome abnormality	41	15	19	7	44.1
Abnormality or variation in maternal origin	19	12 **	1 ***	6	92.3

* Confirmation of confined placental mosaicism (CPM) through placenta biopsy at birth; ** numerical value indicating whether or not the genetic results of the mother herself matched; *** the size of the maternal partial deletion is about 5 M, which is difficult to confirm in general karyotyping. In addition, the fetus obtained an arr (1–23)×2 result through CMA, and no partial chromosomal deletions were found.

## Data Availability

The datasets presented in this article are not readily available to protect patient confidentiality and privacy. All data generated or analyzed during this study are included in this article.

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
