# Peer review of "Cell-Free Fetal DNA Screening Analysis in Korean Pregnant Women: Six Years of Experience and a Retrospective Study of 9327 Patients Analyzed from 2017 to 2022"

_jpm, 2023, doi:10.3390/jpm13101468_

Round 1
Reviewer 1 Report
The quality both of figure 1 and 2 show vague for publication. It needed replace by the clear figures.
Author Response
Please see the attachment.
Thank you
JE Park

Reviewer 2 Report
Dear Author/s,
The manuscript titled "Cell-free fetal DNA screening analysis in Korean pregnant women: Six years of experience and a retrospective study of 932 patients analyzed from 2017 to 2022" is a valuable contribution to the field. With a few minor revisions, the clarity and impact of your findings can be further enhanced.
Suggestive changes
1. Abstract: Page 1, Line 17: Remove the word ‘testing’ out of two.
2. Statistical analysis, Page 4, Line 140: ‘The data were analyzed using R’, Define this sentence.
3. Figure 1, page 4, Line 154: Elaborate about the figure.
4. Table 7, page 9: Correct the spelling of ‘True’.
5. Page 9, Line 297: ‘when the average maternal weight is 160 kg’… recheck the data about average weight because this is impossible.
6. Page 10, Line 337-338: these findings need to be discussed comparatively with other studies.
7. Page 10, Line 347-349: add a citation for this statement.
8. Page 11, Line 358: ‘The no-call group showed a higher prevalence (2.7%’…. in this sentence erase the bracket.
Minor corrections are required
Author Response

(The authors gave the same response as above.)
